# Design principles for engineering bacteria to maximise chemical production from batch cultures

Ahmad A. Mannan[1,3], Alexander P. S. Darlington [2,3] ✉, Reiko J. Tanaka [1] & Declan G. Bates[2] ✉

Bacteria can be engineered to manufacture chemicals, but it is unclear how to optimally engineer a single cell to maximise production performance from batch cultures. Moreover, the performance of engineered production pathways is affected by competition for the host's native resources. Here, using a 'host-aware' computational framework which captures competition for both metabolic and gene expression resources, we uncover design principles for engineering the expression of host and production enzymes at the cell level which maximise volumetric productivity and yield from batch cultures. However, this does not break the fundamental growth-synthesis trade-off which limits production performance. We show that engineering genetic circuits to switch cells to a high synthesis-low growth state after first growing to a large population can further improve performance. By analysing different circuit topologies, we show that highest performance is achieved by circuits that inhibit host metabolism to redirect it to product synthesis. Our results should facilitate construction of microbial cell factories with high and efficient production capabilities.

Bacteria can be engineered to produce chemical products of interest, where production is realised from a population of the engineered bacteria that either simultaneously grows and produces (one-stage production) or that first grows then switches to product synthesis (two-stage production). While chemical production is often measured at the population level, genetic/metabolic engineering itself is done at the level of the single cell. Typically, engineered strains are selected based on their growth and/or synthesis rate. For one-stage bioprocesses, expression of some host enzymes is knocked out to couple and re-balance growth and synthesis[1,2], while for two-stage bioprocesses genetic circuits can be engineered into the host cell that activates maximum product synthesis upon induction after a growth phase[3–7].

However, during lab engineering, it remains unclear how to select growth and/or synthesis rates from production strain candidates which achieve maximum performance at the level of the whole batch culture.

Here, we define the culture-level performance metrics as volumetric productivity and product yield (Equation (3), 'Methods' 'Batch culture model and evaluating production from a cell and from culture'). Volumetric productivity and yield are key culture-level production performance metrics as maximising productivity maximises production rates and maximising yield minimises wastage of substrate consumed, resulting in reduced capital and operational costs for efficient bioproduction[8]. Traditionally, titre and rates are also evaluated to assess the econometrics of batch production, but volumetric productivity is a more useful measure because it specifies how much product is produced per unit of reactor volume per unit of time, and so is directly linked to the required capital investments in the production plant[9].

Furthermore, engineering efforts are complicated by the competition between product synthesis and growth due to the host's limited native cellular resources[10]. In bacterial cells, the expression and

[1]Department of Bioengineering, Imperial College London, London, UK. [2]Warwick Integrative Synthetic Biology Centre, School of Engineering, University of Warwick, Coventry, UK. [3]These authors contributed equally: Ahmad A. Mannan, Alexander P. S. Darlington. ✉e-mail: a.darlington.1@warwick.ac.uk; d.bates@warwick.ac.uk

regulation of enzymes and proteins is optimally tuned by evolution so that the demand for ribosomes and the synthesis of all cellular resources (enzymes, metabolites, energy) are in proportions that achieve high cell growth[11,12]. However, when cells are engineered with heterologous product synthesis enzymes and genetic circuits this optimal balance is perturbed. Expressing heterologous genes utilise the cell's translational resources and cellular metabolites are consumed by the product synthesis pathways. This also often attenuates host growth, thus creating feedback that can indirectly affect both gene circuit function[13–15] and product synthesis[10]. How to engineer a cell that will maximise productivity and yield from the culture, in light of the complications arising from competition for limited cellular resources, is still an open question.

Here, we take a computational approach to address this question, using a 'host-aware' modelling framework[14] and multiobjective optimisation methods. We develop a multi-scale mechanistic mathematical model by augmenting a cell model, capturing the dynamics of cell growth, a simple metabolism, host enzyme and ribosome biosynthesis, heterologous gene expression and product synthesis, with additional expressions capturing the dynamics of population growth, nutrient consumption, and production in batch culture. We then apply multiobjective optimisation methods to reveal how to optimally engineer enzyme expression levels and the gene circuitry controlling them at the single cell level, in the presence of the modelled resource competition, to maximise volumetric productivity and product yield.

## Results

### Selecting strains to maximise culture production performance

Engineering microbial cell factories can involve creating a library of strains in the lab through genetic variations that affect expression or activity of the synthesis pathway enzymes and host cell metabolic enzymes that will re-direct metabolic flux to synthesis, and then selecting strains with desirable characteristics. A change in expression can be effected by altering promoter sequence (RNA polymerase binding strength) or ribosome binding site, and a change in enzyme activity can be effected by encoding for different enzyme variants of different sizes and turnover rates. Production strains are sought whose population will achieve the maximum volumetric productivity, while also delivering high yield, to ensure a cost-effective chemical production from batch culture. It is laborious and costly to test the productivity and yield of many different production strains in the lab, and so often experimental efforts focus on determining a strain's specific growth and synthesis rates[16,17]. However, it is unclear what growth and synthesis rates should be sought to select a strain that achieves maximum productivity and yield from the batch culture. In particular, here we investigate: (i) how do strains with maximum growth and synthesis rates perform at the culture level, (ii) how should we tune expression of host and synthesis pathway enzymes to achieve maximum productivity and yield, and (iii) what are the resulting growth and synthesis rates of strains with maximum culture level performance. Here, we consider transcription rates as tuning dials in the search for optimal protein expression values that will maximise growth and synthesis or productivity and yield. One could also tune ribosome binding sites, which will effect similar changes in protein abundance.

The first question defines a multiobjective optimisation problem, which involves exploring how to scale the transcription rates of a host cell enzyme $E$ and synthesis pathway enzymes $E_p$, $T_p$ to maximise product synthesis ($r_{Tp}$) and host growth rates ($\lambda$) (Fig. 1a). In the context of this work, $E$ represents the host enzyme in a growth limiting pathway, at the branch point between the cell's native metabolism and the heterologous synthesis pathway—a typical situation encountered when engineering bacteria for synthesis of non-native products[18]. Mathematically, this problem is defined in Equation (4) ('Methods'

'Multiobjective optimisation problems for maximising production'), with the scaling coefficients to the transcription rate of the respective enzyme's gene denoted $sTX_E$, $sTX_{E_p}$, $sTX_{T_p}$. Growth ($\lambda$) and synthesis ($r_{Tp}$) rates are calculated from simulations of the host-aware model of the single cell described in Supplementary Note SN1, i.e. the model ignoring dynamics of the batch culture described in Equation (1). We found a Pareto front of optimal scaling values to the transcription expression rates that exhibit a trade-off between growth rate and synthesis rate (green crosses Fig. 1a). To reveal the culture-level performance that each of these 'optimal strains' achieves, we simulated batch culture (i.e. the model including culture-level dynamics described in Equation (1)) and calculated volumetric productivity and yield (Fig. 1b, top). This revealed a range of volumetric productivity and yield performances (green crosses Fig. 1d), including suboptimal designs that give a lower yield for a given productivity (left of the volumetric productivity peak Fig. 1c). For instance, taking three example designs, varying from those with low growth-high synthesis (Fig. 1a, black circle), medium growth-medium synthesis (Fig. 1a, grey circle), and high growth-low synthesis (Fig. 1a, light-grey circle), we find that their simulations for batch culture (Fig. 1c) show performances with high yield-low productivity, medium yield-maximum productivity and low yield-low productivity, respectively (Fig. 1b).

The second question also defines a multiobjective optimisation problem, which involves exploring how to scale the transcription rates that will maximise productivity and yield. This problem is mathematically defined in Eq. (5) ('Methods' 'Multiobjective optimisation problems for maximising production'), where productivity and yield are calculated by Eq. (3) from batch culture simulations ('Methods' 'Batch culture model and evaluating production from a cell and from culture'). We found a Pareto front of optimal designs exhibiting a trade-off between productivity and yield (red-yellow dots Fig. 1b)—all these points lie to the right of the maximum of the productivity-yield trade-off curve seen when maximising growth and synthesis (green crosses, Fig. 1b). We then calculated the growth and synthesis of these optimal designs that maximise productivity and yield using the cell-only model (yellow-red dots, Fig. 1a). Though all these designs fall on the same Pareto front as designs maximising growth and synthesis (green crosses, Fig. 1a), they all favour higher synthesis (Fig. 1a).

These results suggest that it is possible to select single strains that will achieve close to the maximum productivity and yield simply by selecting those strains based on their growth and synthesis rates. In particular, strains with slow growth but fast synthesis rates can be selected to achieve high yields (yellow dots, Fig. 1a, b), whereas strains with comparatively faster growth but slower synthesis rates can be selected to achieve high productivity (red dot, Fig. 1a, b). However, there is an optimal sacrifice in growth rate (0.019 min⁻¹, Fig. 1a) required to achieve the maximum productivity (red dot, Fig. 1b). This suggests that current engineering strategies that focus on obtaining high cell growth may not find strains with high yield and productivity during scale-up. Simulations suggest that strains selected with very high growth rates consume most of the substrate for biomass rather than product, making productivity low (Fig. 1c, bottom). Conversely, strains selected with too low a growth rate but high synthesis also achieve low productivity, as smaller populations take much longer to make as much product (Fig. 1c, top).

The key design principles for engineering strains for high synthesis but low growth (for high yield) are high expression of synthesis enzymes $E_p$, $T_p$ but low expression of host enzyme $E$ (Fig. 1d, e), whereas to engineer strains for lower synthesis and higher growth (i.e. for high productivity), the opposite design principles apply (Fig. 1d,e). It is important to note that engineering for maximum productivity may be challenging because it is difficult to identify how low an expression of synthesis enzymes $E_p$, $T_p$ and how high an expression of the host enzyme $E$, should be sought to achieve maximum productivity, (Fig. 1c).

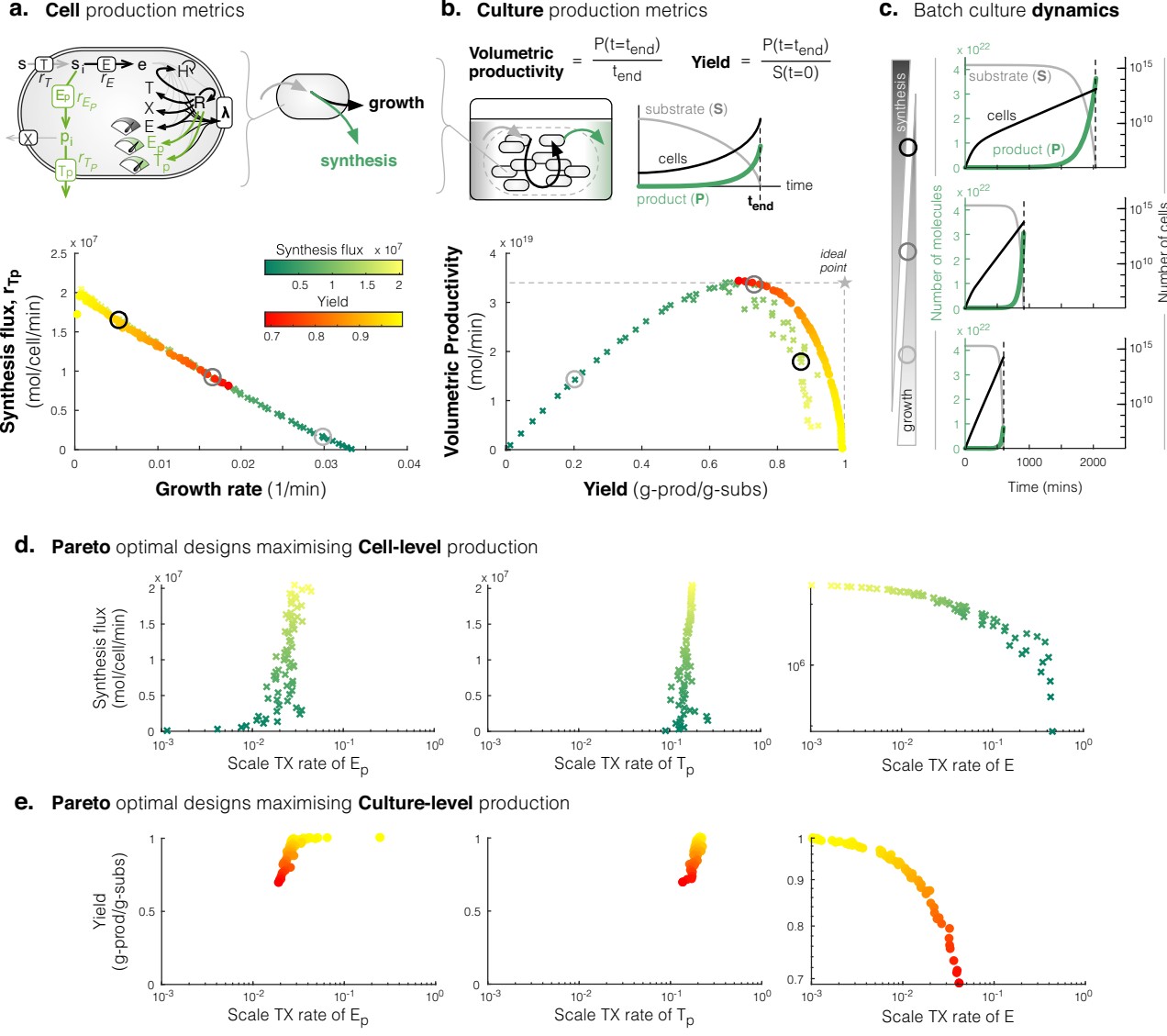

**Fig. 1 | Selecting single strains with lower growth but higher synthesis can maximise productivity and yields from the culture. a** Green-to-yellow crosses show the Pareto front from maximising synthesis and growth rates of the single cell. The calculated productivity and yields of these optimal designs are shown by the green-to-yellow crosses in (**b**). Schematic (top) highlights the three enzyme transcription rates explored over for engineering the production strain, and the cell production metrics. **b** Red-to-yellow dots show the Pareto front from maximising productivity and yield from a batch culture. The calculated synthesis and growth rates of these optimal designs are shown by the red-to-yellow dots in (**a**). Schematic (top) illustrates culture-level production metrics. **c** Time-course simulation of batch culture of three designs selected from fronts in (**a**) and (**b**), spanning designs with high synthesis-low growth to high growth-low synthesis. **d, e** Plots of the optimal scaling on transcription rates of each of the Pareto optimal 'designs' when maximising on synthesis and growth (**d**) or volumetric productivity and yield (**e**).

## Designing genetic circuits to maximise culture performance

So far, we have considered a one-stage production process and noted that a sacrifice in growth is needed to reach the highest productivity and yield. This means smaller populations of cells synthesising products, which inherently constrains culture performance. We then investigated if higher performance could be reached if we enact a two-stage production strategy[19,20] - allowing cells to first grow maximally to a large population and then inducing all cells to the high synthesis-low growth behaviour, at some optimal switch time. Inducible genetic circuits have been re-purposed to enact the growth-synthesis switch[3,4,7,21], but the optimal circuit design and corresponding induction time that maximises productivity and yield remain an open question. We pose this as a multiobjective optimisation problem (defined in Eq. (6)), exploring the following to find their optimal values: induction time ($\tau$), enzyme expression rates, and the topology and parameters of a genetic circuit in which a

transcription factor (TF) is chemically induced at some optimal time to activate the expression of product synthesis enzymes $E_p$, $T_p$[7,21], or deactivate the expression of host metabolic enzyme $E$[4], or both[3,6] (Fig. 2a, circuit in red). We found that indeed dynamic control can achieve even higher productivity compared to only tuning the constitutive expression of the enzymes (Fig. 2a, all curves compared to blue curves).

The circuit which is induced to deactivate host enzyme $E$ and activate synthesis pathway enzymes $E_p$ and $T_p$, at some optimal time, is predicted to give the highest volumetric productivity and yield (Fig. 2a, red curve). Analysis of the Pareto optimal parameters (designs) of this circuit revealed that to achieve high productivity we should engineer production strains in which the activated expression of pathway enzyme $E_p$ is low and is activated by inducer later in the batch culture (Fig. 2b, Supplementary Fig. 16). Conversely, to achieve high yields (at lower productivity), we should engineer the host enzyme $E$ to be

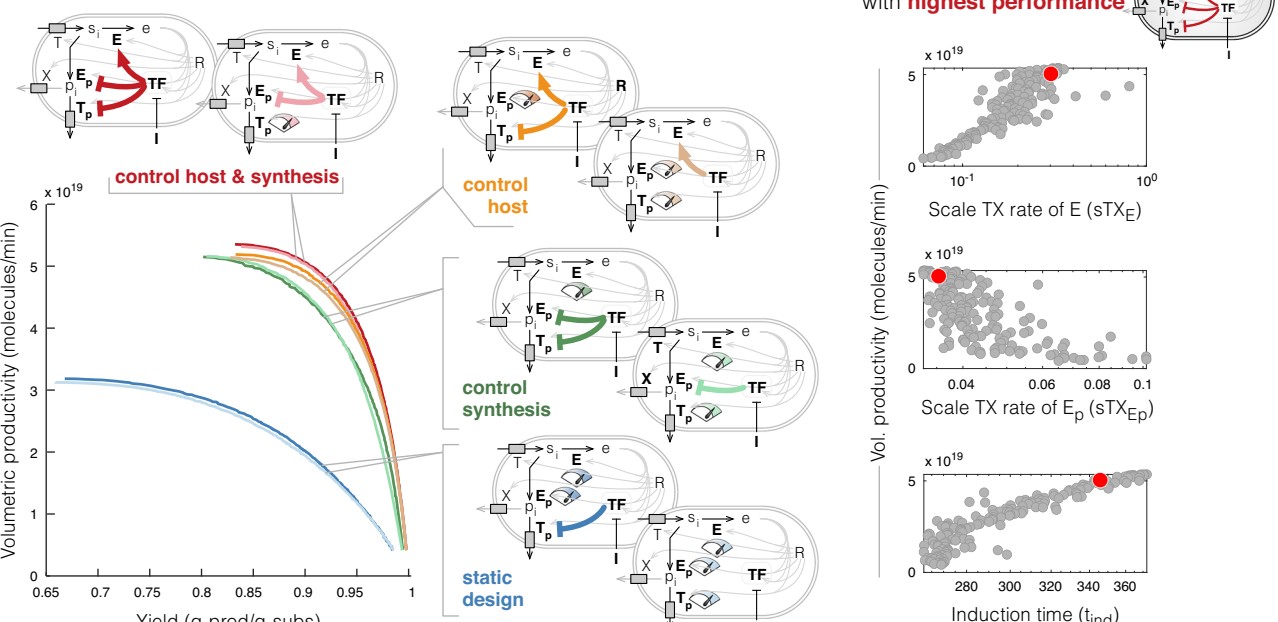

**a.** Pareto front of **optimal dynamic control circuits**

**b.** Optimal params of circuit with **highest performance**

control host & synthesis

control host

control synthesis

static design

Scale TX rate of E (sTX$_E$)

Scale TX rate of E$_p$ (sTX$_{Ep}$)

Induction time (t$_{ind}$)

Volumetric productivity (molecules/min)

Vol. productivity (molecules/min)

Yield (g-prod/g-subs)

**Fig. 2 | Pareto front of optimal dynamic control circuits. a** Pareto fronts of optimal designs for each circuit topology shown. Schematics indicate the circuit topology of the dynamic control by TF and the dials indicate the enzymes whose constitutive expression was tuned. **b** Scatter plots of optimal values of three key parameters which suggest tuning to increase or decrease their values will shift circuit performance to achieve higher productivity but at a cost to yield, or higher yields at a cost to productivity.

expressed at low levels before induction and increase the inducer-activated expression of $E_p$, but activate it earlier compared to designs for high productivity (Fig. 2b, Supplementary Fig. 16).

Comparing the Pareto fronts between all eight circuit topologies, we found that the highest productivity and yields are achieved by circuits with a single common motif—inducer-deactivation of the expression of the host enzyme $E$ (Fig. 2a, motifs at top). Circuits without this motif achieved lower productivity and yields (Fig. 2a). This suggests that forcibly redirecting metabolic flux from growth to the product precursor $s_i$ is necessary to maximise production performance. If we remove control on the synthesis pathway enzymes from the dual-control circuit (Fig. 2a, orange compared to red circuits) the achievable productivity and synthesis are not significantly affected, suggesting activating expression of synthesis enzymes is unnecessary. Our simulations of the dynamics showed that reducing translational precursor $e$, by inducing a deactivated expression of $E$ only, for instance, drives an increase in the translation of all enzymes, i.e. product synthesis enzymes ($E_p$, $T_p$) and nutrient transporter ($T$) (see Supplementary Note SN4). Thus deactivation of host enzyme $E$ indirectly causes the expression of synthesis enzymes to increase, creating an effective regulatory response similar to the dual-control circuit (see Supplementary Notes SN4, SN5). The indirect regulatory increase in expression is observed for any constitutively expressed enzymes, in any of the topologies (Supplementary Note SN5). As a design principle, therefore, the simpler 'one-sided' switch design to deactivate host enzymes may be sufficient for achieving near-optimal performance of the dual-control switch. Moreover, we found that the performance of this 'one-sided' switch compared to the dual-control is generally robust to increases in the burden of the synthesis pathway, i.e. increases in the size and turnover rate of the synthesis pathway enzyme (Supplementary Fig. 15, yellow vs red curves). However, if synthesis enzymes are strongly burdensome, i.e. have very slow turnover or are very large (many codons) compared to native enzymes, then performance of dual-control circuits is superior (Supplementary Fig. 15).

The circuit designs can be simplified further. Focusing on the circuit topology that can achieve the greatest performance, i.e. where TF activates host enzyme $E$ and inhibits synthesis pathway enzymes $E_p$ and $T_p$, we found that removing control on the expression of the synthesis pathway exporter $T_p$, and instead tuning its constitutive expression, does not change the Pareto front (Fig. 2a), nor the design rules (Supplementary Fig. 4) or requirement for inducing later for higher productivity (Supplementary Fig. 16). Varying the activity (i.e. turnover rate—maximum substrate to product conversions per enzyme per min, $k_{cat}$) and size of the $T_p$ shows that this design principle holds for most biologically feasible parameters, with inducer-activation of $T_p$ only useful when the enzyme is especially slow or large (Supplementary Fig. 16). Another key principle that helps to simplify circuit design, is that TF autoregulation (positive or negative) is unnecessary, as it does not affect productivity or yield (Supplementary Fig. 6c), or design rules (Supplementary Fig. 6c), or even the robustness of circuit performance to variations in circuit parameters (Supplementary Fig. 6b), although a higher TF expression rate should be engineered if it is to be engineered with positive autoregulation (Supplementary Fig. 6c).

To achieve optimal performance, we found that most parameters of each control circuit should be tuned to within a narrow range (Supplementary Fig. 4). This raises the concern of how robust performance (productivity and yield) is to variation in these parameters. Assessing the robustness of the Pareto optimal circuit design at the 'knee' of the convex Pareto front (as described in 'Methods' 'Assessing performance robustness to parameter variations' section), for each circuit topology, we found little difference in the robustness between almost all circuit topologies (Supplementary Fig. 5)—no one circuit topology grants greater robustness, or is easier to engineer, than another. However, it is interesting to note that the circuit in which the inducer activates expression of only synthesis pathway enzymes $E_p$, $T_p$, showed a greater loss in performance to parameter perturbations than other circuits (Supplementary Fig. 5), suggesting it is harder to engineer for maximum

**a.** Pareto front of **optimal dynamic control circuits**

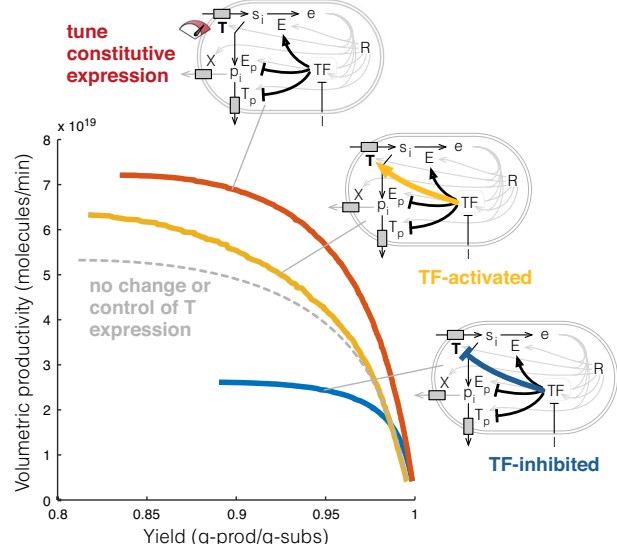

**b.** Optimal params of circuit with **highest performance**

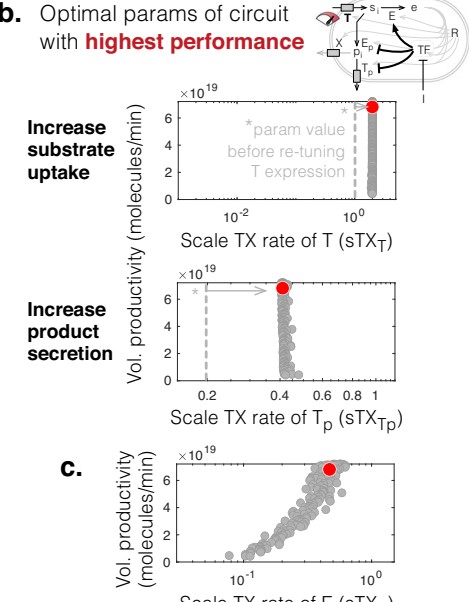

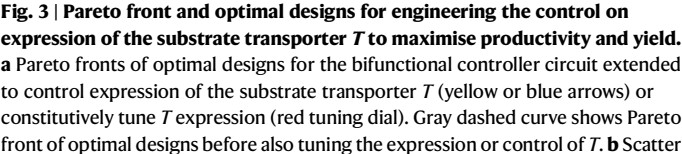

**Fig. 3 | Pareto front and optimal designs for engineering the control on expression of the substrate transporter $T$ to maximise productivity and yield.** **a** Pareto fronts of optimal designs for the bifunctional controller circuit extended to control expression of the substrate transporter $T$ (yellow or blue arrows) or constitutively tune $T$ expression (red tuning dial). Gray dashed curve shows Pareto front of optimal designs before also tuning the expression or control of $T$. **b** Scatter plots of the Pareto optimal values of two key parameters of the circuit with highest performance, which suggests to increase constitutive expression of the substrate and product transporters 2-fold, compared to no tuning (dashed vertical lines). **c** Pareto optimal values of the parameter scaling the transcription rate of host enzyme $E$ for the circuit with highest performance.

performance, but also that it may lose performance quickly due to evolution and/or genetic drift's retuning of its parameters. This is particularly surprising given that this circuit topology is the one most commonly constructed in the lab[7,21].

We also investigated how performance is affected by variations in protein degradation. 93% of proteins are stable, with 64% stable and 29% only slowly degraded (with average half-life = 200 min)[22]. Since these proteins were found to be enriched for annotations (gene ontology) related to metabolism and growth[22], we initially assume that protein degradation rates in our model are negligible. However, these are reported for exponentially growing cells, and so protein degradation may not be negligible after reducing growth upon switching to the production phase. Moreover, there may be rare cases where the heterologous production pathway may utilise enzymes which are unstable, for example, approximately 6% of all cellular proteins are unstable with half-life < 5 h[22,23]. Therefore we updated our model to include protein degradation rates with half-life values of 24, 15, 10, 7 and 3 hours, assuming constant degradation that is independent of cell division rate[24]. We found that decay rates <0.0012 1/min do not alter the shape of the Pareto front, but as decay rate increases, both volumetric productivity and yield fall, with significant losses for decay rates greater than 0.0017 1/min (a protein half-life less than 7 h) (Supplementary Fig. 9). This loss in performance was similar irrespective of whether the production strain is engineered with constitutively tuned expressions (Supplementary Fig. 9a) or control circuits (Supplementary Fig. 9b).

### Increasing expression of host transporters can further increase performance

Constraints on nutrient import can severely limit the growth and production capabilities of the host cell. Interestingly, for bacteria like *E. coli*, it could be physiologically possible to accommodate more transporters on its cell surface[25]. However, if the transporter's constitutive expression is increased, it is unclear if the burden of this

additional expression on limited cell resources would outweigh the benefits of improved nutrient import. For example, an increased expression of transporter could impair growth and reduce synthesis to slow production and cause a drop in productivity, or conversely increase product precursor thus increasing both growth and production. Also, considering control on transporter expression, if its expression is activated with product synthesis, its synthesis could compete with production. Conversely, if transporter expression is inhibited on induction of product synthesis, this could increase product synthesis flux but may deplete the precursor to reduce growth and production. Here, we investigated how re-tuning or regulating (TF activation or repression) the expression of a known nutrient transporter $T$ would impact productivity and yield of the highest performing circuit topology (TF activation of $E$, repression of $E_p$ and $T_p$, and no TF autoregulation).

We found that re-tuning the constitutive expression of the nutrient transporter $T$ nearly doubles productivity (Fig. 3a). The Pareto optimal designs of the circuit suggest that the key principle to increasing productivity and yield is to increase the constitutive expression of nutrient transporter $T$ and product transporter $T_p$ (increase host input and output) by 2-fold (Fig. 3b grey dots, Supplementary Fig. 7a), compared to no tuning of $T$ expression (Fig. 3b dashed line, Supplementary Fig. 4a). Moreover, the key design principle that will enable the tuning of the performance of this circuit from highest yield to highest productivity is to increase the expression of the host enzyme $E$ (Fig. 3c). We found that TF-based deactivation of $T$ after induction could also slightly enhance productivity (Fig. 3a, yellow vs dashed curve), but at the cost of poorer robustness to variations in design parameters compared to the circuit with higher constitutive expression of $T$ (Supplementary Fig. 8).

Altogether, these results suggest that we can engineer for higher production performance by re-tuning the constitutive expression rate of $T$, and although this increases resource burden, the impact is minor and does not require ameliorating with dynamic TF-based control.

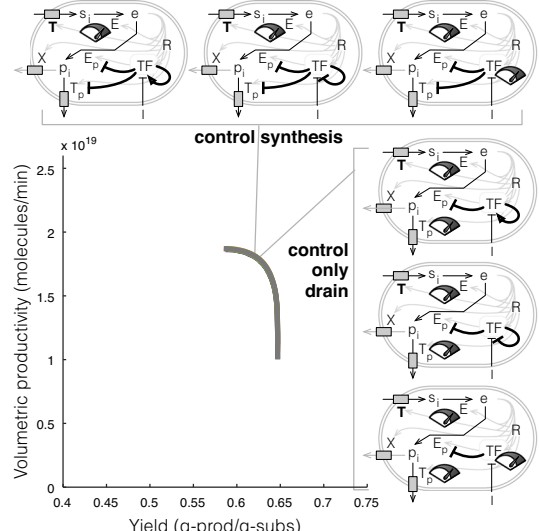

**a.** Pareto fronts of **dynamic control circuits**

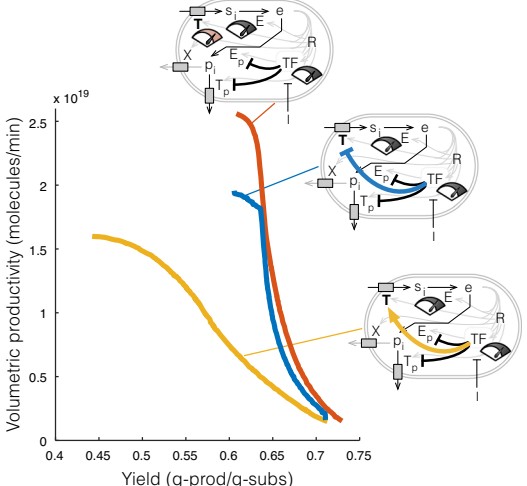

**b.** Pareto fronts of **circuits with extended control on nutrient transporter T**

**Fig. 4 | Top performing dynamic control circuits for when the product of interest directly drains translation precursors. a** Pareto fronts of the top six performing dynamic control circuits. Their Pareto fronts are overlapping. **b** Pareto fronts of the circuit in which TF de-represses expression of pathway enzymes after induction and TF is constitutively expressed, is extended to control expression of nutrient transporter $T$.

## Different design principles are needed to maximise production from translation precursor

We now investigate how the above design principles for engineering control circuits differ if synthesis of the chemical product of interest directly drains translation precursor (denoted $e$ in our model). Examples of such products include the natural sunscreen alternative shinorine (a mycosporine-like amino acid) that drains serine[26], the antimicrobial violacein that drains tryptophan[27], peptides sought as antibiotic alternatives called lantibiotics from staphylococci[28], and derivatives of coumaric acid such as resveratrol[29] and caffeine[30] that drain phenylalanine and tyrosine.

To explore circuit designs where the expression of the host enzyme $E$ and pathway enzymes $E_p$ and $T_p$ can be constitutive or repressed before induction, and where the TF itself may be constitutively expressed or under positive or negative autoregulation (24 distinct topologies), we solve a multiobjective optimisation problem (defined in Eq. (6)) to determined the Pareto optimal circuit designs that achieve maximum productivity and yield. We found that only 6/24 circuit topologies could be engineered to achieve the highest performance (Fig. 4a), the remaining circuits achieved suboptimal performance (Supplementary Fig. 10).

A common motif in the 6 top performing circuits is the constitutive expression of host enzyme $E$ and repression of synthesis pathway enzyme $E_p$ (Fig. 4a). The design principles of these 6 circuits were also very similar: low expression of host enzyme $E$ and product transporter $T_p$, and high expression of synthesis enzyme $E_p$ (Supplementary Fig. 11). Interestingly, performance is predicted to be unaffected by whether or not the production transporter $T_p$ and TF expressions are also placed under control of the TF (Fig. 4a). Also, the 6 circuits were found to have similar robustness to perturbations in circuit parameters (Supplementary Fig. 12). These results suggest there is some flexibility in how to design genetic circuits for high productivity and yield of products that drain translation precursors.

Now considering only the highest performing circuit, where expression of synthesis pathway enzymes $E_p$ and $T_p$ are repressed by the TF, but the constitutive expression of the TF and host enzymes $E$ and $T$ can be tuned, we found that doubling the constitutive expression of $T$ could increase productivity and yield (Fig. 4b). However, this required a re-design of production strains, including doubling the expression of host enzyme $E$, product transporter $T_p$ and TF, weakening repression of synthesis pathway enzymes $E_p$ and $T_p$, and inducing the switch earlier (Supplementary Fig. 13), compared to the circuit before re-tuning $T$ expression (Supplementary Fig. 11). Moreover, the higher performance from increasing constitutive expression of $T$ comes at the cost of poorer robustness of productivity and yield to parameter perturbations, compared to designs with TF-control on $T$ expression (Supplementary Fig. 14).

Altogether, the above results have shown that the design principles for engineering circuits that achieve maximum productivity and yield of a product that drains translational precursor $e$ are different to those for circuits that maximise production of products draining intermediate host cell metabolites.

## Discussion

In this study, we elucidated the principles for how to engineer and control enzyme expression in a single cell to achieve maximum productivity and yield from batch culture. Expression of a heterologous production pathway will compete for both host cell ribosomes and metabolites, so we used a 'host-aware' computational framework to model cell growth and production in the face of these competitions for cellular resources. We found that one can select for strains, which differ in expression levels of host and pathway synthesis enzymes, to achieve near to maximum yield and productivity by selecting strains on maximum growth and synthesis. In particular, strains that are predicted to achieve near maximum yield are those with lowest growth but highest synthesis rates, and strains predicted to achieve near maximum productivity, albeit at a slight cost to yield, are those with lower synthesis rates but higher growth rates. However, strains with even higher growth rates achieve lower productivity and yields, making such strains sub optimal choices. These results suggest that although it is infeasible to evaluate many strains in the lab on productivity and yields, strains that provide close to maximum productivity and yield can be identified simply by selecting strains on maximum growth and synthesis, which are feasible to test in the lab. However, an optimal level of sacrifice in growth is required to achieve maximum productivity, as too high a growth rate means that most of the substrate is consumed for biomass synthesis, while too low a growth rate means small populations take longer to make a given amount of product. Moreover, we found that maximising on growth and synthesis gave an unexpectedly

linear trade-off between the two objectives. This is likely because of the simplicity of the model's representation of metabolism - with metabolic flux only able to go in two directions: growth or synthesis. Nevertheless, our model describes a typical case in which product synthesis competes for a growth-driving precursor or metabolite from central carbon metabolism[18].

The sacrifice in growth necessary to achieve the highest productivity fundamentally limits performance. We found that using genetic circuits to enact two-stage production by inducing the activation of the host's low growth-high synthesis state significantly increases performance. In particular, a dual-control circuit that induces the transcription factor (TF) to both deactivate expression of the host enzyme $E$ and activate expression of synthesis pathway enzymes $E_p$, $T_p$, after allowing cells to grow to a large population, more than doubled productivity, compared to simply re-tuning constitutive expression of all enzymes (one-stage production). This production improvement suggests that the cost of consuming some of the feedstock to first make many cell factories before activating synthesis at lower growths is worth the investment.

Moreover, although the dual-control circuit achieved the highest productivity and yield, we found that performance was not significantly affected if we simplified the circuit design and removed control of expression of the synthesis enzymes $E_p$, $T_p$, similar to our previous work on a much simpler system[31]. This is because competition for cellular resources creates non-regulatory feedbacks that lead to an increased expression of the product synthesis and nutrient uptake enzymes, effectively achieving a similar functionality to the dual-control circuit. We found that switching off expression of host enzyme $E$ reduced translational precursor $e$, which causes a greater loss of ribosomal transcripts compared to transcripts of enzymes. This not only reduced ribosome numbers and cell growth, but the proportionally higher number of enzyme transcripts results in more enzymes being translated, fundamentally shifting the cell proteome to an enzyme-dominated profile[31]. More synthesis and uptake enzymes enabled faster nutrient uptake and product synthesis. These results suggest that the key design principle to enabling highest production performance is to induce the redirection of metabolic flux into the production pathway by increasing synthesis of product precursor, using a simpler circuit topology than dual-control. Indirect activation of the synthesis pathway should then emerge from resource competition. Although redirecting host cell metabolism from growth to production is a known strategy[6,32], in this study we show for the first time that highest productivity and yield capabilities should be achievable with circuits that redirect flux to product precursor, not by those circuits only focused on activating expression of production pathway enzymes.

Another key design principle uncovered in this work is that production performance is unaffected by TF autoregulation. TF autoregulation has been demonstrated to affect the speed of expression response[33], speed or slow recovery from expression from a regulon[34], and increase robustness of switch memory[35]. We speculate that one should be able to augment the growth-production genetic circuit switches with TF autoregulation to take advantage of these functions without affecting production performance or design of the host. For example, production robustness can be increased by engineering in positive autoregulation of the TF, which slows down the ability of the circuit to revert back to the non-production state[35]. Productivity can be increased further by increasing the constitutive expression of the nutrient transporter (e.g. $T$), i.e. there is no need to place it under control. Recent estimates suggest that bacteria like *E. coli* can easily accommodate more nutrient transporters on their cell surface[25], making this a feasible strategy to help realise highest performance capabilities.

A key result from this study is the necessity to redirect the host's metabolic flux to production to achieve maximum production performance. However, how to identify the host enzymes whose deactivation will redirect metabolic flux to the product synthesis pathway remains an open question. Genome-scale computational modelling will be a critical tool to address this, with examples recently appearing in the literature of models being used to elucidate the best control point[32], and a computational tool being developed to identify these so-called 'metabolic valves'[36]. We envision that these tools will help guide the construction of bi-functional inducible dynamic control circuits in the lab. It is, however, important to consider the implications of the system-wide control of metabolic enzymes to redirect flux to production. We expect that repressing the expression of many host enzymes may create metabolic imbalances that result in suboptimal production, similar to those observed in KO production genotypes[16]. We envision the development of new computational tools that should enable the prediction and identification of these metabolic bottlenecks, which should then also be placed under control to overcome the bottlenecks and hopefully achieve close to optimal production. Moreover, we speculate that the design principles of the genetic circuits elucidated in this study could be augmented and layered with genetic circuits that also act to optimize flux through the pathway itself[5,6,37].

A final consideration is how the econometrics of production will affect the choice of how to engineer the production strain. Our results confirm that productivity and yield are opposing objectives, but the choice of engineering for higher productivity or higher yield will depend on the value of the chemical product of interest and the feedstock used in the batch culture. Econometrically, higher yield is desired where the value of the product is low compared to the culture feedstock, whereas higher productivity is desired where product value is high compared to the feedstock. Interestingly, our results show that relatively few parameters govern the productivity-yield trade-off, ensuring in vitro engineering can be carried out with minimal component screening. The tuning dial for higher productivity or higher yields is scaling up or down, respectively, either the transcriptional expression rate of host enzyme $E$ in the case where product synthesis drains an intermediate host metabolite (e.g. $s_i$), or the induction time in the case where product synthesis directly drains transcriptional precursors or amino acids (e.g. $e$).

The above design principles and insights should help guide the engineering of dynamic control circuits of relatively low complexity, in a wide range of host bacteria, to realise high volume but efficient production of high-value chemicals.

## Methods

### The 'host-aware' model of a cell

We developed a mechanistic mathematical model (system of ordinary differential equations) of a host microbial cell factory, accounting for cellular resource competition[13,14], and its structure is detailed in Supplementary Note SN1. In brief, the model describes the dynamics of the expression of a heterologous product synthesis pathway ($E_p$, Supplementary Fig. 1a) and key cellular processes of the host, including the metabolite and ribosome-mediated syntheses of enzymes/proteins and ribosomes, metabolic reactions converting external substrate into internal metabolites ($s_i$), protein synthesis precursors ($e$) and the chemical product of interest ($p_i$), and cell growth (Supplementary Fig. 1a).

### Batch culture model and evaluating production from a cell and from culture

**The batch culture model.** The model is fully described in Supplementary Note SN1.4, but, in brief, the batch culture model is defined as:

$$
\begin{aligned}
\frac{dB(t)}{dt} &= \lambda(t) \cdot B(t), \\
\frac{dS_x(t)}{dt} &= r_T(T(t)) \cdot B(t), \\
\frac{dP_x(t)}{dt} &= r_{T_p}(T_p(t)) \cdot B(t), \\
\frac{dI_x(t)}{dt} &= r_{\text{uInd}}(t) \cdot B(t),
\end{aligned}
\tag{1}
$$

where $B(t)$, $S_x(t)$, $P_x(t)$, $I_x(t)$ model the dynamics of the population biomass, and the external substrate, product, and inducer concentrations, respectively. The specific rates of growth $\lambda(t)$, substrate uptake $r_T(T(t))$, product secretion $r_{T_p}(T_p(t))$, and inducer uptake $r_{\text{uInd}}(t)$ are not fixed parameter values but dynamical variables, calculated from the host-aware cell model, embedded within this batch culture model.

**Evaluating production from the cell genotype.** We define production from a single cell with specific parameters, which we refer to as a production genotype, at the mid-exponential growth rate $\lambda$ and the export rate $r_{T_p}$ which are both defined in SN1 as:

$$\lambda = (1/M_0) \cdot \gamma(e) \cdot \sum_x (c_x), \quad r_{T_p} = v_{T_p}(p_i, T_p) + v_{\text{export}}(p_i, X), \quad (2)$$

Briefly, $\lambda$ is calculated within the cell model as a consequence of global protein production rate, where $\gamma(e)$ is the peptide elongation rate and $\sum_x (c_x)$ is the total number of translating ribosomes, in complex with the mRNA of the $x$th enzyme. $r_{T_p}$ is a Michaelis-Menten function transporting the intracellular product $p_i$ to the external media, using a non-specific host transporter $X$ and specific transporter $T_p$ (Fig. 1a). $e$, $c_x$, $p_i$ and $p_{Tp}$ are calculated as described in Supplementary Note SN1.

**Evaluating production from batch culture.**

$$vP = \frac{P(t = t_{\text{end}})}{t_{\text{end}}}, \quad pY = \frac{P(t = t_{\text{end}})}{S(t = 0)}, \quad (3)$$

where $P(t = t_{\text{end}})$ is the product titre at the end of the simulated batch culture, the point when substrate runs out ($S(t) = 0$), and $S(t = 0)$ is the initial substrate concentration in the culture.

**Multiobjective optimisation problems for maximising production**

We formulate and solve three main multiobjective optimisation problems in this study:

- we first considered maximising volumetric productivity and yield from one-stage batch culture production of a population of a strain whose enzyme expressions are re-tuned (not inducibly controlled) ('Results' 'Selecting strains to maximise culture production performance');
- but also considered maximising product synthesis ($v_{E_p}$) and Growth ($\lambda$) from a single cell, to uncover and elucidate the principles for how to engineer a single cell to achieve maximum productivity and yield from batch culture ('Results' 'Selecting strains to maximise culture production performance');
- and then we considered maximising productivity and yield from two-stage batch culture production of a population of a strain whose enzyme expressions are inducibly controlled by a genetic circuit ('Results' 'Designing genetic circuits to maximise culture performance').

We define each multiobjective optimisation problem below.

**Maximising product synthesis and growth from a cell.** $\lambda$ and $r_{T_p}$ represent the growth and synthesis rates of the single cell characterised by a set of fixed parameters, which can represent a production genotype. We thus define these rates as the *production genotype objectives*. We want to maximise these objectives, searching over and identifying the set of optimal enzyme transcription rates ($T_x(\text{enz})$; i.e. genotype designs). We did this by solving the multiobjective

optimisation problem:

$$\max_{\hat{T}_x(E), \hat{T}_x(E_p), \hat{T}_x(T_p)} [r_{T_p}, \lambda]$$

where $\quad \hat{T}_x(\text{enz}) = \text{sTX}_{\text{enz}} \cdot T_x(\text{enz}), \quad \text{for enzymes (enz)}: E, E_p, T_p,$

given $\quad 0 < \text{sTX}_{E_p} \leq 1, \quad 10^{-3} \leq \text{sTX}_{T_p} \leq 1, \quad 10^{-3} \leq \text{sTX}_E \leq 1,$

$$(4)$$

where $\text{sTX}_{E_p}$, $\text{sTX}_{T_p}$, $\text{sTX}_E$ denote the constants we explore the values of in the optimisation search, which scale the transcription rate of the respective product synthesis enzymes $E_p$, $T_p$ and host enzyme $E$, namely $T_x(E_p)$, $T_x(T_p)$, $T_x(E)$ (see Supplementary Note SN1). In the host-aware model, we assume the turnover rate kinetics of enzymes $E_p$, $T_p$ are the same as the host enzyme $E$ and nutrient transporter $T$.

**Maximising culture-level production from one-stage production.** We were interested in maximising the volumetric productivity (vP) and product yield (pY), as calculated from the batch culture simulation. We thus define vP and pY as the *culture production objectives*, calculated as in Eq. (3). These metrics capture the production capacity of the whole biotechnological process, including population growth and culture time. We want to maximise vP and pY, again searching over and identifying the set of optimal enzyme transcription rates ($T_x(\text{enz})$). We did this by solving the multiobjective optimisation problem:

$$\max_{\hat{T}_x(E), \hat{T}_x(E_p), \hat{T}_x(T_p)} [vP, pY]$$

where $\quad \hat{T}_x(\text{enz}) = \text{sTX}_{\text{enz}} \cdot T_x(\text{enz}), \quad \text{for enzymes (enz)}: E, E_p, T_p,$

given $\quad 0 < \text{sTX}_{E_p} \leq 1, \quad 10^{-3} \leq \text{sTX}_{T_p} \leq 1, \quad 10^{-3} \leq \text{sTX}_E \leq 1,$

$$(5)$$

where the scaling factors $\text{sTX}_{\text{enz}}$ are as defined in the previous paragraph.

**Maximising culture-level production from two-stage production.** We are also interested in maximising vP and pY from two-stage batch culture production. We enact and simulate two-stage production by chemically inducing a TF-based genetic circuit to alter the expression of the host enzyme $E$ and synthesis pathway enzymes $E_p$, $T_p$, at some batch culture time. We want to maximise vP and pY:

$$\max_{g_{\text{ct}}, \text{sTX}_{\text{enz}}, K_{\text{enz}}, \tau} \left[ \frac{vP}{vP^{\text{max}}}, \frac{pY}{pY^{\text{max}}} \right], \quad (6)$$

and searched over genetic circuit designs, as characterised by

(i) the circuit topology, defined in vector $g_{\text{ct}}$:

$$g_{\text{ct}} = [g_T, g_E, g_{E_p}, g_{T_p}, g_{\text{TF}}], \quad (7)$$

whose elements $g_{\text{enz}} \in \{-1, 0, 1\}$ indicate if the TF is inhibiting, not controlling, or activating the expression of the respective enzyme, respectively;

(ii) scaling the enzyme transcription rate ($T_x(enz)$):

$$\hat{T}_x(\text{enz}) = \text{sTX}_{\text{enz}} \cdot T_x(\text{enz}), \quad \text{for enzymes (enz)}: T, E, E_p, T_p, \text{TF}, \quad (8)$$

where we explore over the scaling factor, denoted as sTX$_{enz}$, within the following range:

$$\text{where} \quad 10^{-3} \leq \text{sTX}_{\text{enz}} \leq 2,$$
$$\text{where sTX}_{\text{enz}} \text{ denotes sTX}_T, \text{sTX}_E, \text{sTX}_{E_p}, \text{sTX}_{T_p}, \text{sTX}_{\text{TF}}; \tag{9}$$

(iii) the TF's 'strength' of control on the respective enzyme, denoted $K_{enz}$:

$$\begin{cases} 10^{-6} \leq K_{\text{enz}} \leq 1 & \text{if} \quad g_{\text{enz}} \neq 0, \\ K_{\text{enz}} = 0 & \text{if} \quad g_{\text{enz}} = 0, \end{cases} \quad \text{where } K_{\text{enz}} \text{ denotes } K_T, K_E, K_{E_p}, K_{T_p}, K_{\text{TF}}; \tag{10}$$

and (iv) the induction time ($\tau$):

$$0 \leq \tau \leq 24 * 60. \tag{11}$$

It is important to note that vP and pY objectives are normalised by their respective maximum values (vP$^{max}$, pY$^{max}$), as their absolute values are different by orders of magnitudes and the normalisation helps alleviate bias for optimising changes in the objective with the larger value. Please see Supplementary Note SN3 for details of how we calculated vP$^{max}$ and pY$^{max}$.

In 'Results' section 'Designing genetic circuits to maximise culture performance', we specifically solve the optimisation problems:

(5.1) Solve Equation (6), but explore circuits with TF-control on only host enzyme $E$ and synthesis enzymes $E_p$, $T_p$, which we define with constraints $g_T = 0$, $g_E \in \{0,1\}$, $g_{E_p} \in \{-1, 0\}$, $g_{T_p} \in \{-1, 0\}$, $g_{\text{TF}} = 0$, and sTX$_T = 1$.

(5.2) Solve optimisation problem as defined in (5.1) above, but considering extended genetic circuit with TF-autoregulation, defined by additional constraint $g_{\text{TF}} \in \{-1, 0, 1\}$.

In 'Results' section 'Increasing expression of host transporters can further increase performance', we specifically solve the optimisation problem:

(5.3) Solve optimisation problem as defined in (5.1) above, but considering extended genetic circuit with TF-control extended to host transporter $T$, defined by additional constraint $g_T \in \{-1, 0, 1\}$.

### Assessing performance robustness to parameter variations

Having solved the above problems, we assess how robust the productivity and yield (performance) are to variations in the optimal parameters defining a Pareto optimal design (parameters). For a given circuit topology, we focus on the Pareto optimal design that lies at the 'knee' of its convex Pareto front, i.e. the Pareto optimal design at the 'best trade-off point' between productivity and yield, and randomly vary all the parameters of this design between −20% and +20% of their optimal value. We then re-calculate productivity and yield and measure their change from their values at the optimal parameters. We repeat this 1000 times, find the perimeter of the productivity-yield plane that contains all the values of the productivity and yields after perturbation, and define the robustness measure as the total area of the polygon defined by the perimeter found (e.g. Supplementary Fig. 5b).

### Numerical methods

**Simulations of production dynamics and batch culture.** All simulations were performed in MATLAB using the ODE solver suite function `ode15s`.

**Solving the multiobjective optimisation.** All multiobjective optimisation problems, as defined in 'Methods' 'Multiobjective optimisation problems for maximising production' section above, are solved using the MATLAB Global Optimization Suite function `gamultiobj`.

### Reporting summary

Further information on research design is available in the Nature Portfolio Reporting Summary linked to this article.

## Data availability

Optimisation results generated from this study and presented in the main manuscript are available in the Source Data file. Source data are provided with this paper.

## Code availability

All computational models and analyses were developed in MATLAB 2023b. The authors declare that all MATLAB codes of the models and optimization, for replicating the results presented in the main body, are available at GitHub https://github.com/apsduk/mannan-nat-commun-2024. The code repository is also deposited in Zenodo and citable with doi 10.5281/zenodo.14289807.

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

## Acknowledgements
A.A.M. and R.J.T. are funded by Innovate UK and the European Union EIC Pathfinder (SKINDEV 'Skin microbial devices', 101098826). A.A.M. and D.G.B. were funded by the BBSRC (BB/M017982/1). A.P.S.D. is funded by the Royal Academy of Engineering under their Research Fellowship Scheme (RF/202021/20/270), the EPSRC (EP/Y00342X/1, EP/X039587/1) and the UKRI Technology Mission Fund via the BBSRC (BB/Y007603/1).

## Author contributions
A.A.M., A.P.S.D. and D.G.B. designed the research. A.A.M. and A.P.S.D. developed the models, conducted all computations, and plotted the results. A.A.M., A.P.S.D., R.J.T. and D.G.B. discussed the results and wrote the paper.

## Competing interests
The authors declare no competing interests.
