## [Transparent Peer Review file · Nature Communications]

Design principles for engineering bacteria to maximise chemical production from batch cultures

Corresponding Author: Dr Alexander Darlington

Version 0:

Reviewer comments:

Reviewer #1

(Remarks to the Author)

The manuscript by Mannan and colleagues studies through simulation and multiobjective optimization the design principles for engineering bacteria to maximize chemical production. The study mainly focused on batch cultures. The results that are obtained are largely self-obvious, making difficult to assess how much novelty exists on results that could be expected based on intuition. The dynamic regulation and topologies exploration approach is an interesting asset of the manuscript, as it goes beyond the most common strategies used in metabolic engineering.

However, the conclusion that might be drawn from this work is that no special benefits might be expected by using sophisticated design algorithms like optimal control, compared with more traditional trial-and-error combinatorial screening approaches. The reason for this shortcoming, in this reviewer's opinion is that the authors try to extract general design principles from a simplified case study that does not take into account the complexity and rich variety of bio-based chemical targets and metabolic pathways that are of interest in the field, as well as regulation strategies that can be applied beyond transcriptional regulation.

I have more focused comments, that I detailed below:

Page 2. How often are volumetric productivity and product yield used as culture-level performance metrics? Are there other metrics that should be also taken into account?

Page 3. Not only promoters can be exchanged, often other factors like enzyme variants or ribosome binding sites can have more impact. Why did the authors consider these other strategies?

Page 3. It is not clear to which "host cell enzyme E" is referring, as the host can harbor hundreds of different enzymes, with several of them influencing the synthesis pathway.

Figure 1b. Why the Pareto front in Figure 1b looks almost linear? This is not always the case, as it depends on the synthesis pathway of the product and how it is constrained compared with the metabolic network.

Page 5. How can the authors explain the "sacrifice" in growth rate that is necessary in order to achieve maximum productivity? Is this related to the diverted flux towards product synthesis? In that case, such sacrifice will largely depend on the maximum theoretical yield, with cases showing an important decrease in growth necessary in order to achieve maximum productivity. Please discuss this point in more detail.

Figure 2a. In Figure 2a, how the chemical induction time of the TF is determined? Is this a fixed threshold or was it also considered as a design parameter in the multiobjective optimization?

It is stated that the activate is performed at some optimal time, how this optimal time calculated? Does it change in each context?

Page 6. Redirecting metabolic flux from growth to the product precursor is a known fact, although it depends on each case,

on the precursor's involvement in growth and the achievable titers.

Page 8. The result that the expression of a nutrient transporter would impact productivity and yield seems trivial.

Page 13. In the end, it is not clear what is the actual value of the "design principles", most of them already known, and how they would actually guide a new design beyond providing some intuitive basic principles.

(Remarks on code availability)

The code is only available on request to authors. Moreover, it is written for the proprietary software MATLAB.

Reviewer #2

(Remarks to the Author)

see attached review

(Remarks on code availability)

I was not able to run the code as I don't have matlab. However, code was well documented, including a comprehensive README

Version 1:

Reviewer comments:

Reviewer #2

(Remarks to the Author)

I am happy with the authors' responses to my queries and I believe they have also offered a robust rebuttal to the other reviewer's concerns.

(Remarks on code availability)

Response to Reviewers

Design principles for engineering bacteria to maximise chemical production from batch cultures

Ahmad A. Mannan^{1,†}, Alexander P.S. Darlington^{2,†,*}, Reiko J. Tanaka¹ and Declan G. Bates^{2,*}

¹Department of Bioengineering, Imperial College London, London, SW7 2AZ, UK

²Warwick Integrative Synthetic Biology Centre, School of Engineering, University of Warwick, Coventry, CV4 7AL, UK

[†]equal contribution

*correspondence to a.darlington.1@warwick.ac.uk, d.bates@warwick.ac.uk

Reviewer 1

C1.1 The manuscript by Mannan and colleagues studies through simulation and multiobjective optimization the design principles for engineering bacteria to maximize chemical production. The study mainly focused on batch cultures. The results that are obtained are largely self-obvious, making difficult to assess how much novelty exists on results that could be expected based on intuition.

We thank the reviewer for their comments on our paper. We respectfully disagree with the reviewer's view that our results are intuitive, for the following reasons:

- Our paper contains the first quantitative comparison of the relative performance of optimised one-stage versus two-stage bioprocess strategies using a mechanistic model *that accounts for competition for finite cellular resources*. Accounting for finite cellular resources links together the expression of all proteins modelled in a non-linear way (see Supplementary Figure 2h), making it difficult to intuitively infer how changing the expression of some proteins (through control with gene circuits or constitutively) will affect the expression of other proteins and, in turn, cell growth and production.
- By comparing designs maximising single-cell performance to those maximising culture-level performance, we uncovered how to design the single cell to achieve high performance from the batch culture - a relationship which is complex and non-obvious, making our results of direct relevance for engineering efforts in the lab. We show that optimising expression for maximum synthesis and growth rate *at the single cell level* does not necessarily lead to designs which will maximise the *culture level* performance (i.e. volumetric productivity and yield). However, we also show that optimal designs at the culture

level do fall on part of the Pareto front for single cell metrics - our results therefore provide insights into how lab engineering protocols can be implemented to maximise single cell metrics to achieve high culture productivity and yield.

- Our comparison also reveals that a two-stage strategy based on a dual-control circuit that deactivates expression of host enzymes and activates expression of synthesis pathway enzymes offers large performance improvements over a one-stage approach, a result which is not intuitively obvious *a priori*.
- We subsequently show that direct control of the expression of product synthesis enzymes is not needed, as removing that control, i.e. converting the dual-control circuit to a circuit which controls only host enzymes, achieved similar levels of performance to the dual control circuit. We show that this is due to the presence of non-regulatory feedback loops arising from competition for cellular resources which inherently increases pathway expression when flux is redirected away from growth. This result is both novel and non-intuitive - it could only have been revealed by the inclusion of resource competition constraints allowed for in our modelling framework.
- Another novel and practically useful insight uncovered in this work is that production performance of our control circuits is unaffected by transcription factor autoregulation. Transcription factor autoregulation has been demonstrated to affect the speed of expression response and recovery, and increase robustness of switch memory (see our previous work [1]). It should therefore be possible to augment the proposed growth-production genetic circuit switches with transcription factor autoregulation to take advantage of these functions without affecting production performance or design of the host. For example, production robustness could be increased by engineering in positive autoregulation of the transcription factor, hence slowing down the ability of the circuit to revert back to the non-production state.
- Finally, our study also found that productivity can be increased further by increasing the constitutive expression of the nutrient transporter. The result that there is no need to place it under control is both non-intuitive and unexpected, and we discuss how accounting for limited resources creates non-regulatory feedback that increases transport expression without explicit regulation after switching to low growth. Moreover, recent estimates suggest that bacteria like *E. coli* can easily accommodate more nutrient transporters on their cell surface, making this a feasible strategy to help maximise performance.

C1.2 The dynamic regulation and topologies exploration approach is an interesting asset of the manuscript, as it goes beyond the most common strategies used in metabolic engineering. However, the conclusion that might be drawn from this work is that no special benefits might be expected by using sophisticated design algorithms like optimal control, compared with more traditional trial-and-error combinatorial screening approaches.

Thank you for the reviewer to appreciate our approach. We do believe that there are benefits to be gained from sophisticated design algorithms, which we argue should help to significantly reduce the time spent on traditional trial-and-error combinatorial screening approaches. This is because designs that *cannot* achieve high performance are identified and can thus be avoided, reducing the time and resources spent on lab construction efforts which will fail. For example, our results predict double the production performance through the use of switching control circuits that dynamically regulate the expression of host enzymes and synthesis pathways, as compared to no control. Moreover, design algorithms can also reveal how circuit design can be simplified by enabling one to assess the impact of removing parts of the circuit - helping minimize the difficulty of successfully implementing designs *in vivo*. For example, we found that the performance of single- rather than dual-control switching circuit was similar, so control on production pathway enzymes may not be necessary, except if the enzymes are very large or their turnover is very poor.

As noted in our Discussion, we believe that the identification of host enzymes whose deactivation will redirect metabolic flux to the product synthesis pathway will require the use of sophisticated genome-scale computational modelling and design tools, which will help guide the construction of inducible dynamic control circuits in the lab. We envision the development of new computational tools that should enable the prediction and identification of metabolic bottlenecks, which should then also be placed under control to overcome the bottlenecks and hopefully achieve close to optimal production. Moreover, we speculate that the design principles of the genetic circuits elucidated in this study could be augmented and layered with genetic circuits that also act to optimize flux through the synthesis pathway itself. This is the overall approach we are advocating, and we believe it is a more rational and efficient approach to production strain development compared to “traditional trial-and-error combinatorial screening approaches”.

C1.3 The reason for this shortcoming, in this reviewer’s opinion is that the authors try to extract general design principles from a simplified case study that does not take into account the complexity and rich variety of bio-based chemical targets and metabolic pathways that are of interest in the field, as well as regulation strategies that can be applied beyond transcriptional regulation.

Notwithstanding our disagreement with the reviewer’s characterisation of our conclusions, we think it is a reasonable approach to try to identify useful design principles from general models - we would argue that this is standard practice in the field. Models can also be used to assess robustness of predictions to changes in the model assumptions. For example, we have conducted a sensitivity analysis of our key conclusions to show that our observations are largely robust to pathway burden and kinetics (Supplementary Figure 15 and 16), and protein degradation (Supplementary Figure 9). When considering synthesis of a product of

interest, it is common practice to consider the branch point at the host cell metabolite that is the main product precursor (e.g., [2]). This is what our computational model was constructed to capture. While how to increase the precursor pool of a specific case study will require genome scale modelling and more sophisticated algorithms to identify how to engineer the cell's wider metabolism, the principles elucidated regarding whether engineering efforts should focus on increasing precursor supply or pathway expression, or both, to enable highest performance, are generalisable. While these host growth models will always need to be adapted to the specific industrial strains and pathways, we think that they constitute (a) a useful starting point for the development of more specialised process optimization pipelines, and, more importantly, (b) a demonstration that economically relevant performance improvements are potentially achievable. Our results predict that host-circuit interactions and the associated 'non-regulatory interactions' may enable the engineering of simpler metabolic control circuits where control on host enzymes and growth is needed to ensure high performance, but control on the synthesis pathway is less important because its expression is inherently increased anyway as slowing growth contributes to increasing enzyme pools. Hence, there is no need to explicitly place it under control.

I have more focused comments, that I detailed below:

R1.1 Page 2. How often are volumetric productivity and product yield used as culture-level performance metrics? Are there other metrics that should be also taken into account?

We thank the reviewer for these questions, and apologise for the lack of explanation on the choice of performance metrics. Volumetric productivity and yield are key culture-level production performance metrics as maximising both results in reduced capital and operational costs of bioproduction [3]. Despite the desire to maximise both measures they are often opposing objectives [3] - a result which is also recapitulated in this study. Traditionally, titer, rate and yield are evaluated to assess the econometrics of batch production. We use yield, as it defines the efficiency of the conversion of input substrate to product. Titer and rates are captured by volumetric productivity. As discussed in the recent opinion paper of Konzock and Nielsen (2024) [4], volumetric productivity is a measure of titer normalised by unit time of the batch culture and is a more useful measure than rates for the overall evaluation of a fermentation process. This is because it directly specifies how much product is produced per unit of reactor volume per unit of time, and so is directly linked to the required capital investments in the production plant.

We have now modified the introduction (please see page 2) to expand the explanation for the choice of volumetric productivity and yield as key culture-level performance metrics.

R1.2 Page 3. Not only promoters can be exchanged, often other factors like enzyme variants or ribosome binding sites can have more impact. Why did the authors consider these other strategies?

We agree that when engineering the production strain in the lab, different genetic elements can be manipulated including altering promoter, ribosome binding site (RBS) and encoding for different enzyme variants. Our model is based on the availability of free ribosomes and pools of translation complexes and does not contain details of RNA polymerase competition (which is often deemed small in *E. coli* - see wider discussions such as [5, 6]). The ribosome pool in the model is perturbed by the formation of translation complexes, which can be impacted by varying the mRNA transcription rate (for the synthetic enzyme, w_{Ep}) or ribosome binding site strength (for the synthetic enzyme, b_{Ep}). Simulations show that varying the two parameters leads to the same protein production/growth rate trade-off - i.e. varying either parameter alone (as done in the main text) is sufficient to capture all permissible values of protein production and growth (Response Figure 1). To expand our analysis further would require the modelling framework to be expanded to capture more cellular resources, a key focus of our ongoing research programme. Regarding the impact of enzyme variants, by which we assume the reviewer means encoding for alternative enzymes of the heterologous synthesis pathway, we do indeed investigate how the performance of different circuit topologies is affected by variations in the size (number of codons encoding it, affecting the rate of translation) and turnover rate of synthesis pathway enzymes (see Supplementary Notes 13 and 14). In particular, we found that the result that performance of the dual-switch (control on both host and pathway enzyme expressions) is similar to that of the “one-sided” switch controlling only host enzyme expression holds for large variations in the size and turnover rate of the synthesis pathway enzyme, except when the enzyme is very large, with slow activity. In that case, production performance of the dual-switch is superior (please see page 7).

We have added text on page 3 to note the existence of other tuning dials like ribosome binding sites (RBS) and enzyme variants, and have mentioned that we choose to focus on tuning transcription rates as a means of effecting a change in protein abundances, which could also be done by changing RBS, as seen in the Figure of the simulations below.

R1.3 Page 3. It is not clear to which “host cell enzyme E” is referring, as the host can harbor hundreds of different enzymes, with several of them influencing the synthesis pathway.

We apologise for any lack of unclarity. We have added text on page 4 to clarify what E represents in our model. In this model host cell enzyme E catalyses the conversion of substrate s_i to translational precursor e (representing metabolites such as ATP, GTP and amino acids). In the context of this work E can be considered as the host enzyme in a growth limiting pathway which is at the branch point between the cell’s

Response Figure 1: Simulations of the cell model for a biomanufacturing system where mRNA brith rate (w_{Ep}) and RBS strength (b_{Ep}) are both varied. The panels shows growth rate (λ), steady state protein concentration (p_{Ep} in molecules per cell) and translation rate ($T_L(c_{Ep}, e)$ in peptides produced per min) and concentration (p_{Ep}) versus growth rate (λ). **(A)** A single heterologous protein (i.e. not an enzyme, $k_{cat} = 0$). **(B)** A heterologous enzyme ($k_{cat} = 5800$ molecules per min).

native metabolism and heterologous synthesis pathway. Please also see our response to C1.3 above.

R1.4 Figure 1b. Why the Pareto front in Figure 1b looks almost linear? This is not always the case, as it depends on the synthesis pathway of the product and how it is constrained compared with the metabolic network.

We thank the reviewer for raising this interesting point and agree that the trade-off between growth and synthesis is often not linear for specific pathways. Although our model accounts for limited resources and connects growth and synthesis pathway enzyme expression in a non-linear manner, the almost linear trade-off seen here may be primarily because of the simplicity of the model's representation of metabolism - with metabolic flux only able to go in two directions: growth or synthesis. Nevertheless, the model captures a typical case in which product synthesis competes for a growth-driving precursor or metabolite from central carbon metabolism (King *et al.* (2017) doi:10.1016/j.ymben.2016.12.004), and the result in Figure 1b does recapitulate the established trade-off between growth and synthesis (e.g., Tokuyama *et al.* (2018) doi:10.1002/bit.26568), regardless of the precise shape of the trade-off curve.

We have incorporated a discussion of this important point raised by the reviewer in the Discussion section, see Page 11-12.

R1.5 Page 5. How can the authors explain the “sacrifice” in growth rate that is necessary in order to achieve maximum productivity? Is this related to the diverted flux towards product synthesis? In that case, such sacrifice will largely depend on the maximum theoretical yield, with cases showing an important decrease in growth necessary in order to achieve maximum productivity. Please discuss this point in more detail.

We thank the reviewer for raising this interesting point. All the designs (transcription rates of host enzyme E and synthesis pathway enzymes E_p and T_p) that maximise growth rate and synthesis flux lie on a Pareto front showing a trade-off between growth and synthesis rates, as shown by the crosses in Figure 1b. The growth rates of the optimal designs range from maximum growth at zero synthesis to zero growth at maximum synthesis (dark green to yellow crosses in Figure 1b). The plot of the corresponding productivity and yields of each of these designs is shown in Figure 1c (dark green to yellow crosses). We can see that at highest growth and lowest synthesis rates we have lowest yields and productivity - nothing is being produced. For a small decrease in growth rate, production per cell is improved so total production from a still fast growing population increases total production at a given time in batch culture (i.e., productivity). However, for larger reductions in growth rate, the significant reduction in the total number of cells making product, at the same time point in batch culture as the previous scenario, outweighs the increase in production per cell, resulting in an overall low productivity. Maximum productivity lies in between. This means a minimum sacrifice in growth is needed to sufficiently increase production per cell but also to allow fast enough growth to large enough populations so that the total accumulated product at a given point in batch culture is maximised. We have revised Figure 1 with additional plots of the batch culture dynamics from three different designs, ranging from slow growth-high synthesis to fast growth-low synthesis (Figure 1c). We have also added a more detailed explanation of this point on page 4 and the first paragraph of the Discussion.

R1.6 Figure 2a. In Figure 2a, how the chemical induction time of the TF is determined? Is this a fixed threshold or was it also considered as a design parameter in the multiobjective optimization? It is stated that the activate is performed at some optimal time, how this optimal time calculated? Does it change in each context?

We apologise that this point was not clear in the paper. The induction time, τ , was optimised in the multiobjective optimisation problems, as shown in the mathematical definition of the optimisation problem defined in Equation 6, where the range of times explored over is defined in Equation 10. In Figure 2b, we report that the optimal induction times (of the designs along the Pareto) is correlated with volumetric productivity

- showing that early induction is predicted to lower volumetric productivity. We find differences in induction time between the different topologies tested (see Figure S3) and see that the range of suitable induction times has quite different bounds between these different topologies.

We have rephrased the text in the first paragraph of Results subsection 2.2, page 6, to explicitly mention induction time as a key variable to be optimised.

R1.7 Page 6. Redirecting metabolic flux from growth to the product precursor is a known fact, although it depends on each case, on the precursor's involvement in growth and the achievable titers.

We agree with the reviewer that redirecting host cell metabolism from growth to production is a known strategy that has been used in some experimental studies for small molecule production, e.g., [2, 7]. However, what was unknown is how the choice of either redirecting flux to product precursor (through regulating host enzymes) or increasing production pathway expression, or both, affect the achievable production capability. In our study, we show for the first time that highest productivity and yield capabilities should be achievable with circuits that redirect flux to product precursor, not by those circuits only focused on activating expression of production pathway enzymes.

We have added text into the Discussion, on page 12, to emphasise this result of this study.

R1.8 Page 8. The result that the expression of a nutrient transporter would impact productivity and yield seems trivial.

While it may be intuitive that increasing expression of the transporter would increase productivity, our goal here was to consider how expression of the nutrient transporter *should be regulated* to achieve maximum productivity and yield. We considered this a non-intuitively obvious question for the following reasons:

- If the transporter's constitutive expression is increased, it is unclear if the burden of this additional expression on limited cell resources would outweigh the benefits of improved nutrient import. For example, an increased expression could impair growth and delay the switch to production causing an overall drop in productivity, or the additional expression burden may be outweighed by the increase in growth and product precursor, resulting in an overall increased production.
- If the transporter is activated on induction of product synthesis, its synthesis could compete with expression of the production pathway enzymes. So, whilst substrate import would improve (improve product precursor) the reduction in synthesis enzymes may bottleneck the synthesis pathway flux.
- However, conversely, if the transporter is inhibited on induction of product synthesis, this could increase

product synthesis flux by increasing expression of the production pathway enzymes (due to more cellular resource availability) despite reduced substrate import.

Our results, as depicted in Figure 3, provide answers to these questions, and allow us to conclude that constitutive expression of the nutrient transporter results in the best performance in terms of productivity and yield.

We have incorporated the above discussion in the first paragraph of Results subsection 2.3, page 8-9.

R1.9 Page 13. In the end, it is not clear what is the actual value of the “design principles”, most of them already known, and how they would actually guide a new design beyond providing some intuitive basic principles.

As detailed in our response to comment C1.1, we disagree that the design principles uncovered in this paper are already known or intuitively obvious. The overall point is that it is still unclear what an optimal inducible metabolic control circuit looks like and which of its features (circuit topology) will enable or inherently limit the highest production performance capabilities from batch cultures with the engineered strains. The results of our work could help to improve production capabilities by encouraging the redesign of inducible metabolic control circuits to always incorporate control of the expression of the host cell enzymes to increase product precursor pools. Moreover, uncovering that TF autoregulation does not significantly affect culture production capabilities is also a key design principle that will enable metabolic engineers to capitalise on the ability to tune switch memory, for example, without sacrificing production capability.

R1.10 (Code availability): The code is only available on request to authors. Moreover, it is written for the proprietary software MATLAB.

All ODEs and parameters are given the supplementary material with the intention of transparency and enabling readers/users to recreate/resimulate our work in their chosen software and methods of numerical integration. All our code, developed in MATLAB, was provided with the submission, and it will be published online in a GitHub repository when the paper is published.

Reviewer 2

C2.1 The authors present a computational analysis, using “host-aware” mathematical models, of optimum production strategies for microbial cell factories. The manuscript demonstrates the conflict between max-

imising yield and productivity, and presents the key tunable parameters for favouring each. They go on to demonstrate the improvement that a two-stage, induced production strategy can have on both yield and productivity, identifying the best regulatory topologies, and highlighting areas with diminishing gains. Finally, they demonstrate that a production process that requires “translation precursor” rather than some more upstream substrate, favours a different regulation strategy. The topic of optimal regulatory strategies for bio-production has a broad audience, and systematic, in silico, explorations such as that presented here, are really valuable to the community. The analyses presented here are extensive and the findings are interesting and highlight immediately implementable ideas. I really enjoyed the work but found that there was quite a difficult “step” in understanding to get over in the initial presentation of results that could be eased with a more gentle description of some concepts (as I’ve tried to expand on below). *Alex Fedorec*

Comments and questions on the model:

R2.1 In the model equations presented in SN1.1, during the process of translation ($TL(c_j, e)$) complex c_j sees a loss (ribosome dissociating from mRNA), and ribosome R sees a corresponding gain. But mRNA m_j does not see the corresponding gain. Is this a deliberate choice to model translation as destructive of the mRNA?

We thank the reviewer for raising this typographical error. We have amended the dynamics of the mRNA m_j in SN1.1 to include the release of mRNAs from the translation complex, which was correct in the code but is now corrected in the text. The dynamics are now correctly written in Eq(13) as:

$$\dot{m}_j = \underbrace{T_{X,j}(e) \cdot F(\cdot)}_{\text{Transcription}} - \underbrace{b_j \cdot R \cdot m_j}_{\text{Ribo. binding}} + \underbrace{u_j \cdot c_j}_{\text{Unbinding}} + \underbrace{T_L(c_j, e)}_{\text{Translation}} - \underbrace{(\lambda + \delta_m) \cdot m_j}_{\text{Dilution and decay}} \quad (1)$$

R2.2 There is no protein degradation in the model. I think this is fine under “normal”, exponential growth, circumstances in which dilution from growth dominates cellular protein loss. However, given that the work presented here explicitly models a batch culture, and regulation strategies in which cells are forced into a “starved” state, protein degradation may have an impact on whether a slow production but still growing vs a high production but static strategy is optimum.

We originally assumed that protein degradation rates in our model are negligible as > 93% of proteins are stable (64%) or only slowly degraded (29%, average half life = 200 minutes) [8], making dilution due to growth dominate the dynamics of protein loss. Moreover, these proteins were found to be enriched for annotations (gene ontology) related to metabolism and growth [8]. However, we agree with the reviewer that this may not be the case during the end stages of batch growth and during the production phase. We also note there

may be rare cases where production may utilise enzymes which are unstable (for example, Negar et al. [8], found that approximately 6% of all cellular proteins are unstable with an average half-life of 48 mins and Mosteller et al. [9] found that approximately 5% were unstable (10/189 proteins they tested) with half-life < 5 hrs [9]). Therefore we updated our model to capture the impact of constant protein degradation rate, that is independent of cell division rate, as found by Gupta et al. [10]:

$$\underbrace{\frac{dp_X}{dt} = T_L(c_X, e) - \lambda \cdot p_X}_{\text{Original model with only dilution}} \implies \underbrace{\frac{dp_X}{dt} = T_L(c_X, e) - (\underbrace{\lambda}_{\text{Dilution}} + \underbrace{\delta}_{\text{Decay}}) \cdot p_X}_{\text{Model with decay}} \quad (2)$$

We resolved optimisation problems in the main text assuming decay rates $\delta = [0.0005, 0.0008, 0.0012, 0.0017, 0.0039, 0.0116]$ 1/min (equivalent to protein half lives of 24, 15, 10, 7, 3 and 1 hours respectively). We focused our re-analysis of the system with no circuit/control and tuning constitutive expressions (static design) and on the systems with genetic control circuits with the following topologies: E(+1)Ep(-1)Tp(0), E(0)Ep(-1)Tp(0), E(+1)Ep(0)Tp(0) and E(+1)Ep(-1)Tp(-1), and illustrated the results in Supplementary Figure 9 (also shown below).

Response Figure 2: Effect of protein degradation on maximised production performance capabilities of strains with optimised constitutive expression and production (a) and strains with optimised dynamic control, of four different circuit topologies (b). All proteins are considered to be degraded, including all enzymes, ribosomes, ribosomal complexes and transporters, at five different constant rates: 0.0005, 0.0008, 0.0012, 0.0017, 0.0039 min^{-1} (i.e., half-life of 24, 15, 10, 7, 3 hours).

We find that decay rates < 0.0012 1/min, do not alter the shape of the Pareto front, but as decay rate increases, both volumetric productivity and yield fall (Response Figure 2). The loss in productivity and yield become significant for decay rates greater than 0.0017 1/min (a protein half life less than 7 hours). All four control circuit topologies, including the two ‘one sided’ systems, show similar sensitivities to decay.

We have discussed these results in a new paragraph at the end of Results subsection 2.2, page 8-9, and added the plots of the results in Supplementary Figure 9.

Comments and questions on the presentation:

R2.3 So much of the presented data takes the same form – Pareto fronts on a strain/culture level, and effects of specific parameters on optimised objective values. Because these are so fundamental to the understanding of the paper, I think it would be valuable to spend a bit more time describing the panels in Figure 1. In particular, ensuring the reader understands where the data comes from, and what each of the points represents is crucial. To help with this clarification, it may be useful to pick two or three of the points and show culture and product trajectories over time so a reader can see what, for example, slow growth but high production looks like.

We really appreciate this useful suggestion that has helped improve the manuscript. We have now expanded the text to more clearly describe the panels in Figure 1, and have expanded Figure 1 with additional illustration of the definitions of the cell-level and culture-level performance metrics, and plots of time-course simulations of three selected designs in Figure 1c, highlighted in Figures 1a and 1b, representing designs that achieve low yield-low-productivity, medium yield-max productivity, and high yield-low productivity performances.

We expanded the text in the second paragraph of Results subsection 2.1, page 4, to discuss these three example cases.

R2.4 I think a bit more care should be taken over the presentation of the model scale parameters vs optimised objectives (e.g. Fig 1 d & e). The parameter limits for optimisation are far broader than the x-axis limits on the plots – the optimal parameter values are really quite constrained but the reader can't immediately see that from the plots. I think you could show, maybe just as insets, the data with x-axes limits set to the allowable range in the optimisation. This would also help to demonstrate what I believe to be your main point here, that parameters optimised for single cell objectives will be different from those for culture level objectives.

Thank you for this suggestion. We have chosen to replace Figures 1d and 1e with the x-axis limits of the plots of the optimal parameters set to the allowable range in the optimisation. Please see the revised Figure 1 and its caption.

Further, when presenting this data you only show a subset of the optimised objectives – in fig 1d&e you show synth flux and yield, in later figs you only show productivity. Is there a reason for not showing everything

and is there a reason for the switch from showing yield to showing productivity?

We apologise for omitting the explanation of why we went from showing flux, growth, yield and productivity when investigating the static designs to focusing on productivity when investigating the optimal designs of inducible metabolic control circuits. We believed that it only made sense to focus on the culture level objectives (productivity and yield), as growth and production would ultimately be induced to switch over at some optimal time. Moreover, it was not necessary to plot how the choice of optimal parameter affects both productivity and yield, as they are inherently related on the Pareto front, so a choice of parameters increasing productivity would inherently mean decreases in yield - allowing us to make inferences on both objectives from plotting optimal parameters against only productivity. Finally, we chose to show productivity based on our interest in keeping the focus on production flux rather than efficiency of conversion.

Finally, I think it may be interesting to see this data on a pairs plot to understand if/how these parameters are correlated.

We have made plots between pairs of the optimal parameters shown in Figure 1d and 1e, where points on those plots are coloured by productivity, and added these as supplementary plots - please see Supplementary Figure 1, which we also show here:

Response Figure 3: Correlations between optimal expression rates of constitutively producing strains. Plots of pairs of Pareto optimal expression rates of production strains that maximise growth and synthesis rates (a) or maximise volumetric productivity and yield (b).

Minor comments and corrections:

R2.5 The use of e and ee seem interchangeable in some places, for example page 14 eq 2 and the corresponding text underneath. Can you be more explicit in their definitions and check, where they are being used, that they are used correctly.

We thank the reviewer for noting the typographical error. Both e and ee refer to the same species and we have changed the text to refer to translation precursor as e only.

R2.6 Page 7, “k_cat” – can you give a brief definition for people not familiar.

We have amended the main text to define k_{cat} the first time it is used on page 8.

R2.7 Page 14, “as a the mid-exponential growth” – grammar

Resolved to “... at the mid-exponential growth rate ...”, now on page 15.

R2.8 (Code availability): I was not able to run the code as I don’t have matlab. However, code was well documented, including a comprehensive README

We thank the reviewer for their positive comments on our readme. While we used MATLAB to solve our numerical integration and optimisation problems, all ODEs and parameters are given in the supplementary material with the hope that it allows interested readers to recreate/re-simulate our work in their chosen methods.

References

- [1] A. A. Mannan and D. G. Bates, “Designing an irreversible metabolic switch for scalable induction of microbial chemical production,” *Nature Communications*, vol. 12, no. 3419, pp. 1–11, 2021. Publisher: Springer US.
- [2] I. M. Brockman and K. L. Prather, “Dynamic knockdown of E. coli central metabolism for redirecting fluxes of primary metabolites,” *Metabolic Engineering*, vol. 28, pp. 104–113, 2015. Publisher: Elsevier ISBN: 1096-7184 (Electronic)\r1096-7176 (Linking).
- [3] N. Anesiadis, W. R. Cluett, and R. Mahadevan, “Dynamic metabolic engineering for increasing bioprocess productivity,” *Metabolic Engineering*, vol. 10, no. 5, pp. 255–266, 2008.
- [4] O. Konzock and J. Nielsen, “TRYing to evaluate production costs in microbial biotechnology,” *Trends in Biotechnology*, vol. 0, May 2024. Publisher: Elsevier.
- [5] M. Scott, C. W. Gunderson, E. M. Mateescu, Z. Zhang, and T. Hwa, “Interdependence of Cell Growth and Gene Expression : Origins and Consequences,” *Science*, vol. 330, no. 6007, pp. 1099–1102, 2010.
- [6] A. Y. Weiße, D. A. Oyarzún, V. Danos, and P. S. Swain, “Mechanistic links between cellular trade-offs, gene expression, and growth,” *Proceedings of the National Academy of Sciences*, vol. 112, no. 9, pp. 1038–1047, 2015. ISBN: 0027-8424.
- [7] Y. Yang, Y. Lin, J. Wang, Y. Wu, R. Zhang, M. Cheng, X. Shen, J. Wang, Z. Chen, C. Li, Q. Yuan, and Y. Yan, “Sensor-regulator and RNAi based bifunctional dynamic control network for engineered microbial synthesis,” *Nature Communications*, vol. 9, no. 1, p. 3043, 2018. Publisher: Springer US ISBN: 4146701805.

- [8] N. Nagar, N. Ecker, G. Loewenthal, O. Avram, D. Ben-Meir, D. Biran, E. Ron, and T. Pupko, “Harnessing Machine Learning To Unravel Protein Degradation in Escherichia coli,” *mSystems*, vol. 6, pp. 10.1128/msystems.01296–20, Feb. 2021. Publisher: American Society for Microbiology.
- [9] R. Mosteller, R. Goldstein, and K. Nishimoto, “Metabolism of individual proteins in exponentially growing Escherichia coli.,” *Journal of Biological Chemistry*, vol. 255, pp. 2524–2532, Mar. 1980.
- [10] M. Gupta, A. N. T. Johnson, E. R. Cruz, E. J. Costa, R. L. Guest, S. H.-J. Li, E. M. Hart, T. Nguyen, M. Stadlmeier, B. P. Bratton, T. J. Silhavy, N. S. Wingreen, Z. Gitai, and M. Wühr, “Global protein turnover quantification in Escherichia coli reveals cytoplasmic recycling under nitrogen limitation,” *Nature Communications*, vol. 15, p. 5890, July 2024. Publisher: Nature Publishing Group.

The authors present a computational analysis, using “host-aware” mathematical models, of optimum production strategies for microbial cell factories. The manuscript demonstrates the conflict between maximising yield and productivity, and presents the key tunable parameters for favouring each. They go on to demonstrate the improvement that a two-stage, induced production strategy can have on both yield and productivity, identifying the best regulatory topologies, and highlighting areas with diminishing gains. Finally, they demonstrate that a production process that requires “translation precursor” rather than some more upstream substrate, favours a different regulation strategy.

The topic of optimal regulatory strategies for bio-production has a broad audience, and systematic, in silico, explorations such as that presented here, are really valuable to the community. The analyses presented here are extensive and the findings are interesting and highlight immediately implementable ideas. I really enjoyed the work but found that there was quite a difficult “step” in understanding to get over in the initial presentation of results that could be eased with a more gentle description of some concepts (as I’ve tried to expand on below).

Alex Fedorec

Comments and questions on the model:

1. In the model equations presented in SN1.1, during the process of translation ($TL(c_j, e)$) complex c_j sees a loss (ribosome dissociating from mRNA), and ribosome R sees a corresponding gain. But mRNA m_j does not see the corresponding gain. Is this a deliberate choice to model translation as destructive of the mRNA?
2. There is no protein degradation in the model. I think this is fine under “normal”, exponential growth, circumstances in which dilution from growth dominates cellular protein loss. However, given that the work presented here explicitly models a batch culture, and regulation strategies in which cells are forced into a “starved” state, protein degradation may have an impact on whether a slow production but still growing vs a high production but static strategy is optimum.

Comments and questions on the presentation:

1. So much of the presented data takes the same form – Pareto fronts on a strain/culture level, and effects of specific parameters on optimised objective values. Because these are so fundamental to the understanding of the paper, I think it would be valuable to spend a bit more time describing the panels in Figure 1. In particular, ensuring the reader understands where the data comes from, and what each of the points represents is crucial. To help with this

clarification, it may be useful to pick two or three of the points and show culture and product trajectories over time so a reader can see what, for example, slow growth but high production looks like.

2. I think a bit more care should be taken over the presentation of the model scale parameters vs optimised objectives (e.g. Fig 1 d & e). The parameter limits for optimisation are far broader than the x-axis limits on the plots – the optimal parameter values are really quite constrained but the reader can't immediately see that from the plots. I think you could show, maybe just as insets, the data with x-axes limits set to the allowable range in the optimisation. This would also help to demonstrate what I believe to be your main point here, that parameters optimised for single cell objectives will be different from those for culture level objectives. Further, when presenting this data you only show a subset of the optimised objectives – in fig 1d&e you show synth flux and yield, in later figs you only show productivity. Is there a reason for not showing everything and is there a reason for the switch from showing yield to showing productivity? Finally, I think it may be interesting to see this data on a pairs plot to understand if/how these parameters are correlated.

Minor comments and corrections:

- The use of e and ee seem interchangeable in some places, for example page 14 eq 2 and the corresponding text underneath. Can you be more explicit in their definitions and check, where they are being used, that they are used correctly.
- Page 7, " k_{cat} " – can you give a brief definition for people not familiar.
- Page 14, "as a the mid-exponential growth" – grammar